# The Entire Intestinal Tract Surveillance Using Capsule Endoscopy after Immune Checkpoint Inhibitor Administration: A Prospective Observational Study

**DOI:** 10.3390/diagnostics11030543

**Published:** 2021-03-18

**Authors:** Keitaro Shimozaki, Kenro Hirata, Sara Horie, Akihiko Chida, Kai Tsugaru, Yukie Hayashi, Kenta Kawasaki, Ryoichi Miyanaga, Hideyuki Hayashi, Ryuichi Mizuno, Takeru Funakoshi, Naoki Hosoe, Yasuo Hamamoto, Takanori Kanai

**Affiliations:** 1Division of Gastroenterology and Hepatology, Department of Internal Medicine, Keio University School of Medicine, Tokyo 160-8582, Japan; kshimozaki@keio.jp (K.S.); shorie192@gmail.com (S.H.); akihiko.chida.214@gmail.com (A.C.); kait721@keio.jp (K.T.); yukie_tennis_0929@hotmail.co.jp (Y.H.); ideaal.retter@hotmail.co.jp (K.K.); mimigo2007@gmail.com (R.M.); takagast@z2.keio.jp (T.K.); 2Keio Cancer Center, Keio University School of Medicine, Tokyo 160-8582, Japan; rock-hayashi-pop@rhythm.ocn.ne.jp (H.H.); yashmmt1971@gmail.com (Y.H.); 3Department of Urology, Keio University School of Medicine, Tokyo 160-8582, Japan; mizunor@z7.keio.jp; 4Department of Dermatology, Keio University School of Medicine, Tokyo 160-8582, Japan; takeruf@a8.keio.jp; 5Center for Diagnostic and Therapeutic Endoscopy, Keio University School of Medicine, Tokyo 160-8582, Japan; nhosoe@z5.keio.jp

**Keywords:** capsule endoscopy, enterocolitis, prospective studies, immune checkpoint inhibitors, observational study

## Abstract

Background: Despite the proven efficacy of immune checkpoint inhibitors (ICIs) against various types of malignancies, they have been found to induce immune-related adverse events, such as enterocolitis; however, the clinical features of ICI-induced enterocolitis remain to be sufficiently elucidated, which is significant, considering the importance of early detection in the appropriate management and treatment of ICI-induced enterocolitis. Therefore, the current study aimed to determine the utility of capsule endoscopy as a screening tool for ICI-induced enterocolitis. Methods: This single-center, prospective, observational study was conducted on patients with malignancy who received any ICI between April 2016 and July 2020 at Keio University Hospital. Next, second-generation capsule endoscopy (CCE-2) was performed on day 60 after ICI initiation to explore the entire gastrointestinal tract. Results: Among the 30 patients enrolled herein, 23 underwent CCE-2. Accordingly, a total of 23 findings were observed in 14 (60.8%) patients at any portion of the gastrointestinal tract (7 patients in the colon, 4 patients in the small intestine, 2 patients in both the colon and the small intestine, and 1 patient in the stomach). After capsule endoscopy, 2 patients (8.7%) developed ICI-induced enterocolitis: both had significantly higher Capsule Scoring of Ulcerative Colitis than those who had not developed ICI-induced enterocolitis (*p* = 0.0455). No adverse events related to CCE-2 were observed. Conclusions: CCE-2 might be a safe and useful entire intestinal tract screening method for the early detection of ICI-induced enterocolitis in patients with malignancies.

## 1. Background

Over the past decade, immune checkpoint inhibitors (ICIs) targeting the programmed cell death 1 (PD-1)/programmed-cell death-1 ligand (PD-L1) or cytotoxic T lymphocyte antigen-4 (CTLA-4) pathway have been proven to be remarkably effective against various malignancies [1,2,3,4,5]; however, they have been known to induce a diverse spectrum of specific toxicities that can develop in any organ, referred to as immune-related adverse events (irAEs), which have remained the main obstacle to ICI treatment despite its efficacy [6,7]. Although some studies have reported possible rationales for the development of irAEs caused by PD-1/PD-L1 or CTLA-4 blockade [8,9,10], integral elements of the underlying mechanisms remain to be determined.

With an incidence rate of 7–54%, enterocolitis has been one of the most common and problematic irAEs [3,11,12,13,14]. Previous clinical trials have shown that ICI-induced enterocolitis has a median onset of approximately 8 weeks [15] and could even occur long after ICI discontinuation [16]. Considering that ICI-induced enterocolitis exhibits nonspecific symptoms, such as abdominal pain, fever, diarrhea, or even none at all, distinguishing it from other types of enterocolitis, such as infectious colitis or diarrhea due to the peritoneum metastases, is often difficult [17]. In addition, ICI-induced gastritis is reported in some studies, which is characterized by symptoms, such as nausea, abdominal pain, or melena stool, and shows mucosal edema and erythema [18,19]. Treatments of the ICI-induced enterocolitis include anti-diarrheals, fluids, and electrolyte supplementation for patients with non-severe symptoms [16]. However, patients with severe symptoms should receive systemic corticosteroids or infliximab, an anti-TNFα antibody. Although there is limited data available from prospective studies with a large sample size, Ibraheim et al. reported in the meta-analysis of 39 studies that corticosteroid efficacy was 59% [20]. Additionally, the response to infliximab in patients who are refractory or intolerant to corticosteroids was reported to be 81%, with 1–3 infliximab doses in most cases. Further, the possible efficacy of vedolizumab, an anti-integrin α4β7 antibody with gut-specific immunosuppressive effects, is investigated for the treatment of severe ICI-induced enterocolitis [21].

Colonoscopy, one of the essential examinations for diagnosing ICI-induced enterocolitis, has been recommended in the guidelines for the management of irAEs [16,22]. Repeat colonoscopy can also help in the evaluation of the treatment for ICI-induced enterocolitis [23]. Endoscopic findings of ICI-induced enterocolitis include a mucosa with diffuse ulceration, edema, and luminal bleeding over the entire or often segmental colon [24,25,26]; however, total colonoscopy for patients with severe symptoms is often difficult to perform. Although sigmoidoscopy can be an alternative to endoscopy, some studies have suggested that ICI-induced colitis occurs more often at the proximal than at the distal colon [11,27,28]. Severe ICI-induced enterocolitis has been a major cause for the discontinuation of ICI treatment, despite its efficacy. Indeed, early identification of the signs of ICI-induced enterocolitis would greatly contribute to its appropriate management. Furthermore, total gastrointestinal tract screening among patients receiving ICI treatment might be useful for the early detection of asymptomatic ICI-induced enterocolitis.

Only a handful of studies have investigated the utility of capsule endoscopy as a screening tool for ICI-induced enterocolitis. Otagiri et al. reported in a case report on performing capsule endoscopy for a patient with malignant pleural mesothelioma and revealed enteritis induced by nivolumab [29]. As a surveillance method for detecting colon polyps, capsule endoscopy has been considered useful due to its greater simplicity and lesser invasiveness than colonoscopy [30]. Some studies have reported capsule endoscopy to be an accurate method for detecting colon polyps greater than 10 mm in size, which is clinically significant when considering polyp resection [31]. Recently, second-generation capsule endoscopy (CCE-2) has been widely used for colon cancer screening and colonic polyp surveillance. The diagnostic ability of CCE-2 in detecting polyps > 10 mm is reported to have 85–92.8% sensitivity and 94–99% specificity in comparison with colonoscopy [31,32]. The positive predictive value of 52–89% and a negative predictive value of 92.9–96% have also been reported [33]. With adequate detection of colorectal polyps, capsule endoscopy is considered to be a suitable alternative to colonoscopy. Studies have also shown that this type of capsule endoscopy can be beneficial for pan-enteric surveillance in patients with Crohn’s disease [34,35]. Considering that ICI-induced enterocolitis could occur in any portion of the gastrointestinal tract, capsule endoscopy could be an ideal method for the surveillance of ICI-induced enterocolitis.

Hosoe et al. recently reported a new simple scoring system, named Capsule Scoring of Ulcerative Colitis (CSUC), to evaluate the severity of ulcerative colitis by assessing the severity of the vascular pattern, bleeding, and erosions and ulcers of the mucosa through capsule endoscopy [36]. Considering that ICI-induced enterocolitis and ulcerative colitis have similar endoscopic findings, CSUC could be utilized to assess the endoscopic severity of ICI-induced enterocolitis. Therefore, the current study investigates the clinical utility and safety of colon capsule endoscopy as a pan-enteric screening method for detecting ICI-induced enterocolitis.

## 2. Materials and Methods

### 2.1. Study Design

This single-center, prospective, observational study was conducted on patients with malignancy who received any ICI at the Keio University Hospital. The inclusion criteria were as follows: (1) histologically proven solid cancer, (2) planned to receive any type of ICIs (either monotherapy or combination therapy) or had initiated ICI treatment for 28 days, and (3) age ≥ 20 years. The exclusion criteria were as follows: (1) medical history of abdominal surgeries or small bowel obstruction, (2) diagnosis of Crohn’s disease, (3) known obvious peritoneal metastases confirmed through computed tomography (CT) or the presence of clinical symptoms associated with intestinal stenosis, (4) presence of cardiac pacemakers or other medical electronic devices, and (5) allergies to mosapride, dimethicone, or polyethylene glycol solution.

All enrolled patients were scheduled to undergo CCE-2 (Pillcam^®^ Colon2, Medtronic, Yokneam, Israel) 8 weeks (within a margin of 2 weeks) after the initiation of ICI treatment. The test schedule is presented in Appendix A. After undergoing CCE-2, patients were followed up according to the study protocol. Those who developed symptoms of diarrhea or colitis and were determined by their attending physicians to have needed endoscopic examination during the follow-up period subsequently underwent the corresponding endoscopy, such as esophagogastroduodenoscopy (EGD) and/or colonoscopy and/or balloon-assisted enteroscopy.

This study was approved by the Keio University Hospital Institutional Ethics Committee and was performed in accordance with the Declaration of Helsinki and Ethical Guidelines for Medical and Health Research Involving Human Subjects. Written informed consent was obtained from all patients. This study was registered in the University Hospital Medical Information Network (UMIN000022149).

### 2.2. Assessment

After assessing CCE-2 videos of the entire gastrointestinal tract, three expert endoscopists (RMiy, NH, and YHay) independently scored findings in the colon according to the CSUC. Discrepancies in the assessment of the three experts were resolved through discussions, during which a consensus regarding which score to adopt was reached. CSUC constitutes three factors, namely, vascular pattern (minimum–maximum, 0–3), bleeding (0–2), and erosions and ulcers (0–3), which are evaluated at the proximal (i.e., from the cecum to the splenic flexure) and distal (i.e., from the descending colon to the rectum) colon. The final score is calculated as the sum of the scores for all three factors (0–14).

This study also collected data on baseline characteristics, medication history (proton pump inhibitors (PPIs) and nonsteroidal anti-inflammatory drugs (NSAIDs)), clinical outcomes following ICI treatment (primary tumor location, type of ICI, treatment line, European Cooperative Oncology Group performance status (PS), ICI treatment duration, overall survival (OS), and follow-up period), Lichtiger index [37], CSUC, irAEs, and blood test results (hemoglobin, white blood cell count, and C-reactive protein). IrAEs were defined as events occurring both during ICI treatment and after ICI discontinuation, including pneumonitis, diarrhea/colitis, hepatitis, rash, neurological disorders, or endocrine abnormalities, which were diagnosed as irAEs by the attending physician. The severity of irAEs was graded according to the Common Terminology Criteria for Adverse Events version 4.0. irAEs predefined herein were AEs that appeared to be associated with the mechanisms of action of ICIs. The use of PPIs or NSAIDs was defined as the continuous administration of both medications for 30 days before or after ICI initiation.

### 2.3. Statistical Analysis

To evaluate the patients’ characteristics, summary statistics were constructed by employing frequencies and proportions for categorical data and means and standard deviations (SD) for continuous variables. Patients’ characteristics were then compared using the Chi-Square test or Fisher’s exact test for categorical data and *t*-tests for continuous data as appropriate. The Mann–Whitney U test was used to compare differences in nominal variables between two independent groups. OS was defined as the duration between ICI initiation and death from any cause. Probabilities of survival were estimated using the Kaplan–Meier method, whereas hazard ratios (HRs) were calculated using the Cox proportional hazard model. All *p-*values were based on a two-sided hypothesis, with those <0.05 indicating statistical significance. All statistical analyses were performed using JMP version 14.2.0 (SAS Institute, Cary, NC, USA).

## 3. Results

### 3.1. Patients’ Characteristics

From April 2016 through July 2020, a total of 30 patients were eligible for the present study (Figure 1). Among the 23 patients who underwent CCE-2, 12 had renal cell carcinoma, 6 had gastric cancer, 2 had esophageal cancer, 2 had malignant melanoma, and 1 had microsatellite instability-high solid tumor. Three types of ICIs were administered: nivolumab (20 patients), pembrolizumab (2 patients), and nivolumab plus ipilimumab (1 patient). A total of 17 patients (73.9%) achieved total gastrointestinal tract observation, whereas the remaining 6 patients (26.1%) had poor observational findings in portions of the gastrointestinal tract. The baseline characteristics of the 23 patients who underwent capsule endoscopy are summarized in Table 1.

### 3.2. Positive Capsule Endoscopy Findings

Among the evaluated patients, 14 had a total of 23 gastrointestinal tract findings (positive-finding group), whereas 9 had no endoscopic findings (no-finding group). Patients’ characteristics at baseline were similar in both groups. The Lichtiger index was not statistically different between the patients with capsule endoscopy findings (median, 0; range, 0–1) and those without any findings (median, 0; range, 0–3) (*p* = 0.38). Findings were observed in the colon in 9 patients (39.1%; 9 proximal and/or 4 distal) and in the small intestine in 6 patients (26.1%) (Figure 2), respectively. Moreover, 8 patients (34.8%) had some findings showing multiple lesions across the gastrointestinal tract. Typical CCE-2 findings are summarized in Figure 3. Evaluation of the small intestine revealed that 1 patient had multiple scattered edematous lesions and 4 patients had erosions.

The positive-finding group had a median CSUC of 1 (range, 0–4). The most frequently observed findings included vascular patterns (9 lesions in 7 patients) and ulcer and/or erosions (8 lesions in 5 patients). A list of the CSUC in the positive-finding group is presented in Figure 4. Two patients (noted as * in Figure 2) developed ICI-induced colitis of grade ≥ 2 after CCE-2 surveillance and were administered immunosuppressive treatments. Two patients who developed ICI-induced colitis had two vascular patterns and four ulcers and/or erosions. Patients who did not develop ICI-induced colitis (n = 12) had seven vascular patterns at any lesions and four ulcers and/or erosions. There was a significant difference between the groups in terms of the frequency of ulcers and/or erosions observed during the capsule endoscopy (*p* = 0.0139); however, there was no statistical difference between the groups with regard to the frequency of vascular patterns (*p* = 0.122). Patients with ICI-induced colitis also had a significantly higher CSUC (4 and 3) than the 12 patients who did not develop ICI-induced colitis (*p* = 0.0445).

### 3.3. Summary of the Two Cases Who Developed ICI-Induced Colitis

Case 1

A 67-year-old man diagnosed with metastatic renal cell carcinoma received nivolumab after being refractory to axitinib. However, CCE-2 performed 60 days after nivolumab initiation showed erosions in the proximal colon, with no findings in the small intestine. On day 112, he was admitted to our hospital with grade 3 diarrhea. According to the CCE-2 results, colonoscopy was performed at first, which revealed the loss of vascular marking and an edematous mucosa at the transverse colon (Appendix A). He had no medication history associated with drug-induced colitis. Further, the fecal culture result was negative. The patient was intolerant to prednisolone administration, owing to grade 2 hiccups; hence, infliximab was administered, and the symptoms subsequently improved.

Case 2

A 50-year-old man diagnosed with gastric neuroendocrine carcinoma received pembrolizumab, owing to the high microsatellite instability status confirmed by polymerase chain reaction-based testing. On day 60, CCE-2 was performed, which revealed an obliterated area, erosion in the proximal colon, and moderate erosions in the distal colon. On day 120, the patient developed severe cough and diarrhea. Computed Tomography (CT) revealed bilateral ground-glass opacity in the lungs; however, there were no findings in the gastrointestinal tract. Considering the CCE-2 results obtained on day 60, we performed colonoscopy, which showed slight and sporadic mucosal changes (Appendix A). Nonetheless, stool culture and Clostridium difficile toxin testing results were negative. The symptoms were subsequently evaluated as irAE Grade 2 and subsided following prednisolone administration.

### 3.4. Survival

As of the data cutoff date (31 July 2020), the median follow-up duration for survival was 16.4 months in both the positive-finding and no-finding groups. Kaplan–Meier curves for survival are presented in Appendix A. The positive-finding group did not reach the median OS, whereas the no-finding group had a median OS of 27.0 months (HR, 0.62; 95% confidence interval (CI), 0.14–2.78; log-rank *p* = 0.402).

### 3.5. Safety

No adverse events related to CCE-2 were observed in this study. A total of 7 patients did not undergo CCE-2 owing to the following reasons: 1 patient withdrew consent, 1 patient was admitted because of a femur fracture, and 5 patients exhibited general condition deterioration associated with disease progression (3 with rapid disease progression, 1 with grade 2 constipation owing to the peritoneal metastasis, and 1 with grade 2 fatigue). The characteristics of the aforementioned patients are detailed in Appendix A.

## 4. Discussion

The current single-center prospective study demonstrated that more than 50% of the patients who underwent CCE-2 (60.8%) exhibited some endoscopic findings without any symptoms 8 weeks after ICI initiation. To the best of our knowledge, this has been the first study to evaluate the clinical utility of CCE-2 as a pan-enteric screening tool for the early detection of ICI-induced enterocolitis. Notably, our results suggest that immunotherapy affects the gastrointestinal tract of more patients than previously assumed. In particular, capsule endoscopy showed small intestinal findings in a total of 6 patients (26.1%). Because ICI could induce enteritis as well, careful follow-up of small intestine is also important for patients receiving ICI treatment [38]. However, it is difficult to evaluate the small intestine by endoscopy for all patients receiving ICI treatment, and CT/magnetic resonance imaging (MRI) might not provide adequate information for enteritis. Thus, CCE-2 can be a valuable pan-enteric surveillance method for detecting not only colitis but also enteritis induced by ICIs. In this study, we observed some edematous lesions and erosions in 6 patients. However, these patients did not develop ICI-induced enteritis; thus, the significance of these findings observed by capsule endoscopy remains uncertain.

Among the patients evaluated herein, 2 (8.7%) had developed grade ≥ 2 ICI-induced colitis, revealing an incidence rate comparable to that previously reported in clinical trials [1,2,13,14]. Interestingly, both patients who developed ICI-induced colitis were also asymptomatic 8 weeks following ICI administration—a novel insight into the possible timing of ICI-induced enterocolitis. Moreover, our results indicated that some abnormal gastrointestinal mucosal findings in patients who developed ICI-induced enterocolitis could have developed much earlier than the appearance of symptoms. Although the sample size was limited in this study, the ulcers and/or erosions were significantly more frequently observed in patients who developed ICI-induced colitis than in patients who did not. This result is reasonable considering that the endoscopic findings of ICI-induced colitis reported in some studies that the ulcers and/or erosions might represent early involvement lesion of the ICI-induced colitis [11,25]. Several studies have reported that the actual development of ICI-induced enterocolitis might be related to certain risk factors, such as the type of ICI, duration of ICI treatment, microbiota, or individual patient-specific immunological predisposition [39,40,41,42]. Despite the need for further investigations to determine the importance of early capsule endoscopy, total gastrointestinal tract surveillance after ICI initiation could be of certain importance for the early detection of ICI-induced gastrointestinal toxicity [43]. Furthermore, early interventions against ICI-induced enterocolitis would certainly help deal with enterocolitis before developing more clinically significant problems and allow the safe continuation or reintroduction of ICI treatment.

The present study also suggested that CSUC, which had been developed for evaluating the severity of ulcerative colitis, could be useful for assessing the development of ICI-induced enterocolitis. Matsubayashi et al. reported that a CSUC of ≥1 was significantly associated with the relapse of ulcerative colitis [44]. Despite the limited number of patients included herein, our result showed that those who developed ICI-induced colitis had significantly higher CSUC than those who did not. Considering studies showing that ICI-induced colitis and ulcerative colitis had similar endoscopic findings, CSUC may predict the development of ICI-induced colitis [28,45]. However, given that the correlation between clinical severity and endoscopic severity in ICI-induced colitis remains controversial, further investigations on the relationship between clinical severity of ICI-induced colitis and endoscopic findings are needed. 

Our capsule endoscopy completion rate (73.9%) was relatively lower than that previously reported for other diseases [46,47], perhaps owing to the following reasons. First, given that all patients included herein had cancer, the intestinal peristaltic activity might have been chronically suppressed. However, CCE-2 observation had reached the proximal colon in 20 patients (87.0%). Second, this study used a reduced-volume regimen consisting of 2000 mL of polyethylene glycol (PEG) solution for patients with poor general condition. Considering that the standard protocol for CCE-2 requires 4–6 L of PEG solution [31,47], the preparation might have been insufficient. Although complete observation is crucial, CCE-2 might be sufficient for surveying ICI-induced colitis. Moreover, the addition of sigmoidoscopy for patients with incomplete observations might be a less invasive alternative.

Most importantly, the current study showed that CCE-2 had been safely performed without any examination-related adverse events. However, 7 patients could not undergo CCE-2, and 5 of them had metastatic gastrointestinal cancer with primary disease progression before the examination. This limitation could be explained by two reasons. First, patients with metastatic gastrointestinal cancer could have had peritoneal metastases and gastrointestinal stenosis, which could not have been detected radiographically before enrollment. In this study, a patency test was not performed before capsule endoscopy. Currently, established methods that can completely avoid capsule retention do not exist, and patient’s symptoms (e.g., postprandial pain), known Crohn’s disease, and a history of chronic NSAID use are risk factors for capsule retention; thus, patients with advanced cancer should be considered as having a possibility of capsule retention [48]. Second, considering that ICIs are only approved for later lines of treatment in this population (e.g., second-line treatment for esophageal squamous cell cancer and third-line or later treatment for gastric cancer), the general conditions were unstable. As such, after carefully considering the patient’s general condition, symptoms of diarrhea, constipation, or abdominal pain, and the potential presence of peritoneal metastases undetected by early radiography, we decided to perform CCE-2. Overall, CCE-2 had been safely performed and can be a suitable screening method for those who were eligible for CCE-2.

Our study has some noteworthy limitations. First, only a limited number of patients have been included herein. Therefore, the present study might have been underpowered to detect a larger clinical effect. To help us better understand the utility of total gastrointestinal tract screening by capsule endoscopy, the study may need to include a larger group of patients and include a treatment arm for patients with capsule endoscopy findings that could be compared with the findings of patients who did not receive prophylactic treatment. Second, the follow-up duration was not sufficiently long to observe adverse events and survival, which may have influenced the results of the present study. Third, the pathological findings could not be investigated in the present study. Microscopic colitis is a subtype that has a normal endoscopic presentation along with active inflammation, as noted in a tissue biopsy. Capsule endoscopy cannot detect microscopic colitis or be used to obtain tissue samples. Fourth, the preparation for capsule endoscopy may be difficult for patients with advanced cancer because it requires the intake of over 2000 mL of PEG; thus, the examination burden for these patients was not reduced compared with that for colonoscopy. Fifth, the present study failed to regulate the decision whether to perform esophagogastroduodenoscopy or colonoscopy if capsule endoscopy revealed remarkable findings. Hence, we suggest performing endoscopic examination after capsule endoscopy detects any findings based on the severity of patients’ symptoms in future studies. In addition, three patients were being treated with NSAIDs due to cancer pain at the time the capsule endoscopy was performed, which might have influenced the endoscopic findings. 

## 5. Conclusions

In conclusion, the present study showed that CCE-2 can be a clinically useful pan-enteric screening method for the early detection of ICI-induced enterocolitis. The entire intestinal tract surveillance in patients treated with ICIs might be beneficial for the early detection of and intervention for ICI-induced gastrointestinal toxicity. CCE-2 has great potential for observation of the entire gastrointestinal tract considering that irAEs can occur in any segment. Further investigations in a large cohort are warranted to establish the utility of CCE-2 for screening patients with ICI-induced enterocolitis.

## Figures and Tables

**Figure 1 diagnostics-11-00543-f001:**
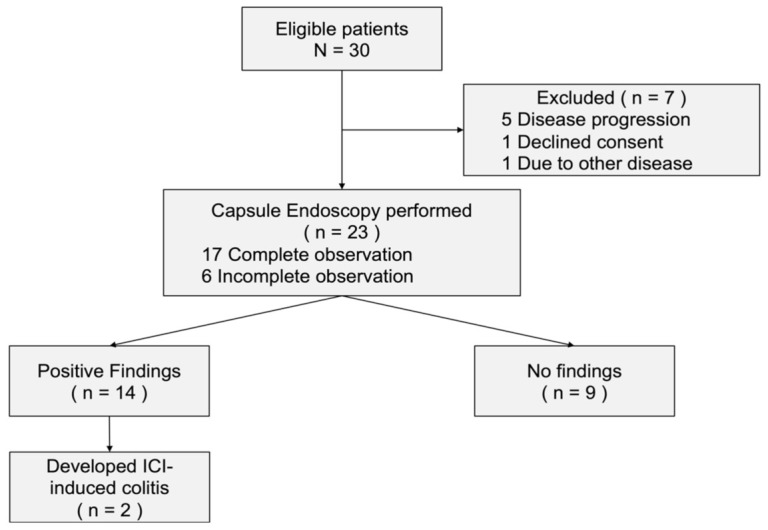
Patient flow chart of the present study. Abbreviation: ICI, immune checkpoint inhibitors.

**Figure 2 diagnostics-11-00543-f002:**
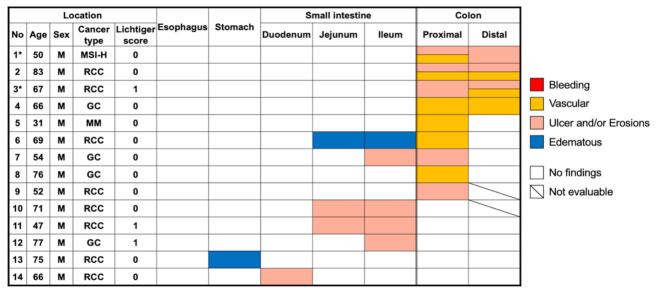
Distribution of positive findings detected following capsule endoscopy. Abbreviations: GC, gastric cancer; M, male; MM, malignant melanoma; MSI-H, microsatellite instability-high; RCC, renal cell carcinoma.

**Figure 3 diagnostics-11-00543-f003:**
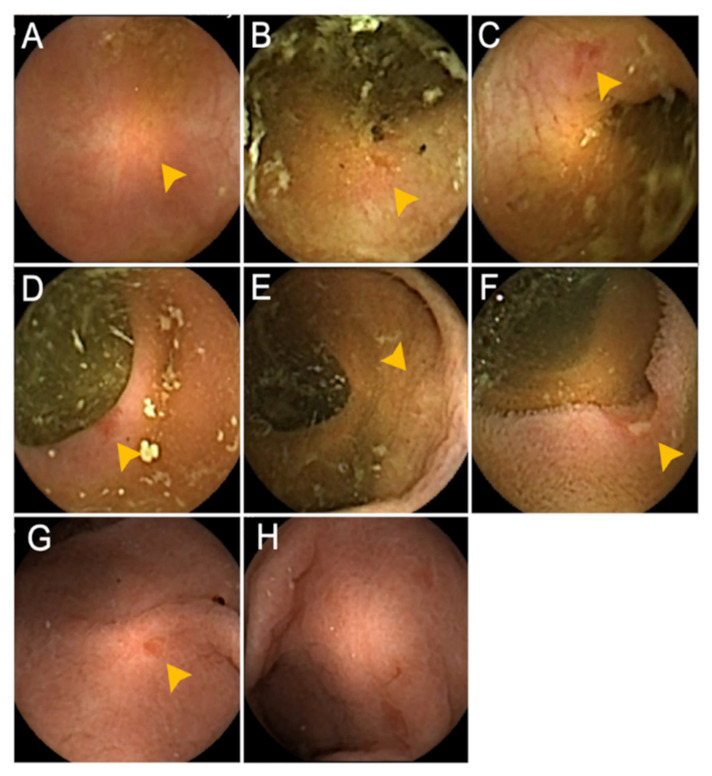
Capsule endoscopy findings detected in the present study. (**A**) Scar at the proximal colon, (**B**) erosion at the proximal colon, (**C**) erosion at the distal colon, (**D**) erosion at the distal colon, (**E**) redness at the proximal colon, (**F**) ulceration at the ileum, (**G**) redness at the stomach, (**H**) edematous mucosa at the stomach.

**Figure 4 diagnostics-11-00543-f004:**
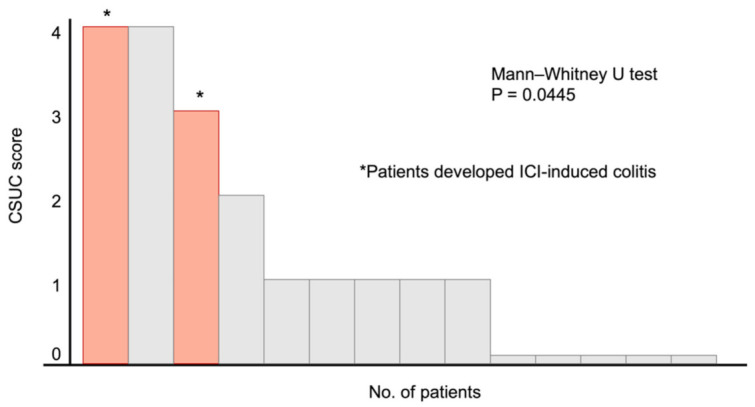
Capsule Scoring of Ulcerative Colitis for the 14 patients who showed positive findings following capsule endoscopy. * Two patients developed immune checkpoint inhibitor-induced colitis after capsule endoscopy. Abbreviations: CSUC, Capsule Scoring of Ulcerative Colitis; ICI, immune checkpoint inhibitors.

**Table 1 diagnostics-11-00543-t001:** Characteristics of patients who underwent the capsule endoscopy.

		Total N = 23	Positive Findings n = 14	No Findings n = 9	*p*-Value
**Median age, range (years)**		67 (31–83)	68 (31–83)	66 (42–79)	0.83
**Male, no (%)**		21 (91%)	14 (100%)	7 (78%)	0.044 *
**BMI, kg/m^2^, range**		19.6 (16–30.4)	21.6 (17.6–30.4)	17.9 (16–28.2)	0.10
**ECOG PS, no (%)**	0	8 (35%)	6 (43%)	2 (22%)	0.30
	1	15 (65%)	8 (57%)	7 (78%)	
**Primary tumor, no (%)**					0.28
Renal cell carcinoma		12 (52%)	8 (57%)	4 (44%)	
Gastric cancer		6 (26%)	4 (29%)	2 (22%)	
Esophageal cancer		2 (9%)	0	2 (22%)	
Malignant melanoma		2 (9%)	1 (7%)	1 (11%)	
MSI-H solid tumor		1 (4%)	1 (7%)	0	
**Immune checkpoint inhibitors, no (%)**					0.32
Nivolumab		20 (87%)	11 (79%)	9 (100%)	
Pembrolizumab		2 (9%)	2 (14%)	0	
Nivolumab plus ipilimumab		1 (4%)	1 (7%)	0	
Median Lichtiger score, range		0 (0–3)	0 (0–1)	0 (0–3)	0.38
**NSAIDs usage, no (%)**		6 (26%)	3 (21%)	3 (33%)	0.37
**PPI usage, no (%)**		8 (35%)	5 (36%)	3 (33%)	0.90
**Baseline laboratory findings**					
WBC, range (/μL)		6400 (3500–13,700)	7000 (3500–13,700)	6200 (4200–7300)	0.20
Hb, range (g/dL)		12.4 (7.7–15.1)	12.6 (7.7–15.1)	12.2 (8.8–14.5)	0.70
CRP, range (mg/dL)		0.25 (0.01–2.8)	0.16 (0.01–2.8)	0.1 (0.01–1.55)	0.42

Abbreviations: BMI, body mass index; CRP, C-reactive protein; ECOG, European cooperative oncology group; Hb, hemoglobin; MSI-H, microsatellite instability-high; NSAIDs, nonsteroidal anti-inflammatory drugs; PPI, proton pump inhibitors; PS, performance status; WBC, white blood cell. One asterisk (*) indicates *p* value smaller than 0.05 (*p* < 0.05).

## Data Availability

The datasets analyzed in this study are available from the corresponding author on reasonable request.

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
