# Peer review of "The Entire Intestinal Tract Surveillance Using Capsule Endoscopy after Immune Checkpoint Inhibitor Administration: A Prospective Observational Study"

_diagnostics, 2021, doi:10.3390/diagnostics11030543_

Round 1

Reviewer 1 Report

I think this is an interesting article on the use of CCE-2 in ICI-induced enterocolitis. It is well-written. I only have a few comments:

  1. The study population (n=23) is small but this has been acknowledged by the authors as a limitation
  2. The routine use of CCE-2 as a screening tool in patients receiving ICI for malignancy is proposed as a potential strategy, yet I am not entirely sure this would be appropriate. Many patients may have a few minor findings (e.g. a couple of erosions, oedema etc) that may never progress to a clinically severe enterocolitis and recommending that they should routinely undergo investigation which requires bowel preparation etc for screening purposes seems to be in my view somewhat aggressive. 
  3. Table 1. How did the authors account for the use of NSAIDs in their study. These can cause erosions etc in the small bowel and colon, so these findings may overlap with ICI-induced enterocolitis. 

Author Response

To Reviewers:

We wish to express our appreciation to the reviewers for their insightful comments on our manuscript (MS. No. diagnostics-1128382). In the following sections, you will find our responses to each of the points and suggestions. We appreciate the time and effort required to provide this guidance.

We hope that these revisions sufficiently improve the manuscript such that it is worthy of publication in Diagnostics. Below, we provide detailed responses to the reviewers’ comments.

REVIEWER #1’s COMMENTS

I think this is an interesting article on the use of CCE-2 in ICI-induced enterocolitis. It is well-written. I only have a few comments:

  1. The study population (n=23) is small but this has been acknowledged by the authors as a limitation.

REPLY: Thank you for the input. We have described the small sample size as one of the limitations of our study.

2.The routine use of CCE-2 as a screening tool in patients receiving ICI for malignancy is proposed as a potential strategy, yet I am not entirely sure this would be appropriate. Many patients may have a few minor findings (e.g. a couple of erosions, oedema etc) that may never progress to a clinically severe enterocolitis and recommending that they should routinely undergo investigation which requires bowel preparation etc for screening purposes seems to be in my view somewhat aggressive.

REPLY: We appreciate your valuable comment. As you indicated, it is difficult to routinely perform whole gastrointestinal surveillance using capsule endoscopy. The incidence of ICI-related enterocolitis is relatively low, and clinicians might not attempt to treat ICI-induced enterocolitis in patients with a subclinical status. Further investigation of the necessity of total gastrointestinal screening in patients treated with ICIs and the appropriate method is warranted. Thus, we corrected the conclusion below to summarize more clearly the utility of whole gastrointestinal tract surveillance and use of capsule endoscopy as a screening tool. Thank you again for the comment.

Conclusions

In conclusion, the present study showed that CCE-2 can be a clinically useful pan-enteric screening method for the early detection of ICI-induced enterocolitis. Whole gastrointestinal tract surveillance in patients treated with ICIs might be beneficial for the early detection of and intervention for ICI-induced gastrointestinal toxicity. CCE-2 has great potential for observation of the entire gastrointestinal tract considering that irAEs can occur in any segment. Further investigations in a large cohort are warranted to establish the utility of CCE-2 for screening patients with ICI-induced enterocolitis.

  1. Table 1. How did the authors account for the use of NSAIDs in their study. These can cause erosions etc in the small bowel and colon, so these findings may overlap with ICI-induced enterocolitis.

REPLY: Thank you for your important suggestion. We agree with the comment that the use of NSAIDs could be the cause of erosions, especially in the small bowel. We have added a description regarding the possible influence of use of NSAIDs on the endoscopic findings as a limitation, as mentioned below. We hope that the reviewer finds this revision an improvement. Thank you again for this valuable comment.

Discussion (lines 317–323)

Third, the pathological findings could not be investigated in the present study. Microscopic colitis is a subtype that has a normal endoscopic presentation along with active inflammation as noted in a tissue biopsy. Capsule endoscopy cannot detect microscopic colitis or be used to obtain tissue samples. In addition, three patients were being treated with NSAIDs due to cancer pain at the time the capsule endoscopy was performed, which might have influenced the endoscopic findings.

Reviewer 2 Report

The authors aimed to clarify the usefulness of capsule endoscopy (CE) as a screening tool for immune checkpoint inhibitor (ICI)-induced enterocolitis. Although only two patients developed ICI-induced “colitis”, and preparation with 2000 mL of polyethylene glycol was required prior to CE, we think it is worthwhile to screen for this disease with less invasive CE. However, there are several concerns about the methodology.

  1. Were there any cases with gastrointestinal symptoms prior to the CE on day 60?

  1. How did you confirm the patency of the gastrointestinal tract prior to performing CE? If CE is retained in a patient with poor general condition, will you perform balloon-assisted enteroscopy to retrieve CE?

  1. The rate of identification of non-specific lesions by CE has been reported to be high; therefore, it is unclear whether the positive findings seen in 14 cases were ICI-induced or not. Positive findings may indicate subclinical IBD; in general, ICI-induced enterocolitis occurs at a high rate in patients with IBD. Did you perform any screening before administering ICI?

  1. In this study, only colonoscopy was performed after the onset of ICI-induced diarrhea; if CE is intended as a pan-enteric screening tool or “whole gastrointestinal tract surveillance”, it should also be performed as a follow-up or after the onset of ICI-induced diarrhea to evaluate small bowel lesions. Otherwise, it is difficult to say, I think, that the clinical utility for small bowel lesions was evaluated.

Author Response

To Reviewers:

We wish to express our appreciation to the reviewers for their insightful comments on our manuscript (MS. No. diagnostics-1128382). In the following sections, you will find our responses to each of the points and suggestions. We appreciate the time and effort required to provide this guidance.

We hope that these revisions sufficiently improve the manuscript such that it is worthy of publication in Diagnostics. Below, we provide detailed responses to the reviewers’ comments.

REVIEWER #2’s COMMENTS

The authors aimed to clarify the usefulness of capsule endoscopy (CE) as a screening tool for immune checkpoint inhibitor (ICI)-induced enterocolitis. Although only two patients developed ICI-induced “colitis”, and preparation with 2000 mL of polyethylene glycol was required prior to CE, we think it is worthwhile to screen for this disease with less invasive CE. However, there are several concerns about the methodology.

  1. Were there any cases with gastrointestinal symptoms prior to the CE on day 60?

REPLY: Thank you for your important comment. Three patients exhibited gastrointestinal symptoms prior to capsule endoscopy. One patient (No. 7 in Figure 2) had endoscopic findings, whereas the others (76-year-old male and 55-year-old-female with gastric cancer ) did not present with any findings. Thank you again for your comment.

  1. How did you confirm the patency of the gastrointestinal tract prior to performing CE? If CE is retained in a patient with poor general condition, will you perform balloon-assisted enteroscopy to retrieve CE?

REPLY: We appreciate your valuable comment. We did not perform capsule endoscopy to confirm patency; instead, patients with known obvious peritoneal metastases confirmed on computed tomography (CT) and those with clinical symptoms associated with intestinal stenosis were excluded from the present study. For the query raised, we had to perform balloon-assisted enteroscopy when the capsule was retained due to peritoneal stenosis. From this point of view, patients suitable for capsule endoscopy should be strictly determined; however, it is often difficult to perform colonoscopy in patients with cancer due to their poor performance status. We believe that capsule endoscopy could reduce the examination burden and contribute to the management of patients treated with ICIs.

  1. The rate of identification of non-specific lesions by CE has been reported to be high; therefore, it is unclear whether the positive findings seen in 14 cases were ICI-induced or not. Positive findings may indicate subclinical IBD; in general, ICI-induced enterocolitis occurs at a high rate in patients with IBD. Did you perform any screening before administering ICI?

REPLY: Thank you for your important suggestion. We did not perform gastrointestinal screening, such as esophagogastroduodenoscopy or colonoscopy. As indicated by you, preexisting IBD predicts the incidence of ICI-related GI toxicities (Hamzah Abu-Sbeih et al. JCO. 2020). However, the incidence of IBD in elderly patients is reported to be low. Considering that cancer patients are usually older, we did not perform GI screening before the initiation of ICI treatment. Thank you again for your valuable comment.

  1. In this study, only colonoscopy was performed after the onset of ICI-induced diarrhea; if CE is intended as a pan-enteric screening tool or “whole gastrointestinal tract surveillance”, it should also be performed as a follow-up or after the onset of ICI-induced diarrhea to evaluate small bowel lesions. Otherwise, it is difficult to say, I think, that the clinical utility for small bowel lesions was evaluated.

REPLY: We appreciate the suggestion provided. In the present study, six patients exhibited findings in the small intestine; however, patients who developed ICI-induced enterocolitis after CCE examination (No. 1 and 3 in Figure 2) did not exhibit any findings in the small intestine. We completely agree with the reviewer that it is difficult to confirm the clinical utility of small bowel lesions from this study. Thus, we have described the following sentences in the discussion section. Thank you again for this important comment.

In this study, we observed some edematous lesions and erosions in six patients. However, these patients did not develop ICI-induced enteritis; thus, the significance of these findings observed by capsule endoscopy remains uncertain (lines 251–253).

Round 2

Reviewer 2 Report

  1. According to your response, you did not perform gastrointestinal testing on three symptomatic patients within 60 days. I do not know when they complained of gastrointestinal symptoms, but please clearly state the reason why you did not perform CE until day 60 despite their complaints of gastrointestinal symptoms.

  1. I also recognize the usefulness of CE to screen the entire gastrointestinal tract, but it requires preparation with 2,000 mL of polyethylene glycol, which does not necessarily reduce the examination burden. In addition, the patency capsule did not used to assess the patency of the gastrointestinal tract, so if the capsule is retained, the patient will have a heavy burden. This point should be mentioned.

  1. According to your response, when your patients developed ICI-induced enterocolitis, did you select the gastrointestinal site to be examined (esophagogastroduodenoscopy, CE, or colonoscopy) according to the results of screening CE examination? If so, the algorithm should be clearly stated.

Author Response

REVIEWER #2’s COMMENTS

  1. According to your response, you did not perform gastrointestinal testing on three symptomatic patients within 60 days. I do not know when they complained of gastrointestinal symptoms, but please clearly state the reason why you did not perform CE until day 60 despite their complaints of gastrointestinal symptoms.

Reply: Thank you for this important remark. Although 3 patients were symptomatic before the examination, their symptoms were mild (mild abdominal pain in one patient and abdominal swelling in two patients), and none of them had symptoms such as bloody diarrhea or severe abdominal pain. Hence, colonoscopy or esophagogastroduodenoscopy was not performed before capsule endoscopy (CE).

  1. I also recognize the usefulness of CE to screen the entire gastrointestinal tract, but it requires preparation with 2,000 mL of polyethylene glycol, which does not necessarily reduce the examination burden. In addition, the patency capsule did not used to assess the patency of the gastrointestinal tract, so if the capsule is retained, the patient will have a heavy burden. This point should be mentioned.

Reply: We completely agree with your suggestion. The preparation for CE might not reduce the examination burden compared with that for colonoscopy. However, we believe that CE could evaluate the entire gastrointestinal tract at once. We considered the examination burden as a limitation of this study. In line with your comment, the possibility of capsule retention should be considered. However, the risk of possible retention is low for patients without Crohn’s disease. Further, Cave et al. described that no established method is currently available to avoid capsule retention, and patient’s symptoms (e.g., postprandial pain), known Crohn’s disease, and a history of chronic nonsteroidal anti-inflammatory drug (NSAID) use might be potential risk factors for capsule retention. Thus, we do not believe that patency capsule is necessary for all patients, and we asked the patients regarding their conditions, medications, or past medical history before performing CE. To explain the possibility of capsule retention and its preventive management, we added the following sentences in the Discussion about safety. Thank you again for your important suggestion.

Discussion

Most importantly, the current study showed that CCE-2 had been safely performed without any examination-related adverse events. However, 7 patients could not undergo CCE-2; 5 of them had metastatic gastrointestinal cancer with primary disease progression before the examination. This limitation could be explained by two reasons. First, patients with metastatic gastrointestinal cancer could have had peritoneal metastases and gastrointestinal stenosis, which could not have been detected radiographically before enrollment. In this study, patency test was not performed before capsule endoscopy. Currently, established methods that can completely avoid capsule retention do not exist, and patient’s symptoms (e.g., postprandial pain), known Crohn’s disease, and a history of chronic NSAID use are risk factors for capsule retention; thus, patients with advanced cancer should be considered as having a possibility of capsule retention[46]. Second, considering that ICIs are only approved for later lines of treatment in this population (e.g., second-line treatment for esophageal squamous cell cancer and third-line or later treatment for gastric cancer), the general conditions were unstable. As such, after carefully considering the patient’s general condition, symptoms of diarrhea, constipation, or abdominal pain, and the potential presence of peritoneal metastases undetected by early radiography, we decided to perform CCE-2. Overall, CCE-2 had been safely performed and can be a suitable screening method for those who were eligible for CCE-2.

Discussion, section of limitations

Fourth, the preparation for capsule endoscopy may be difficult for patients with advanced cancer because it requires the intake of over 2000 mL of PEG; thus, the examination burden for these patients was not reduced compared with that for colonoscopy.

Reference

  1. Cave D, Legnani P, de Franchis R, Lewis BS; ICCE. ICCE consensus for capsule retention. Endoscopy. 2005;37:1065-7. doi: 10.1055/s-2005-870264.

  1. According to your response, when your patients developed ICI-induced enterocolitis, did you select the gastrointestinal site to be examined (esophagogastroduodenoscopy, CE, or colonoscopy) according to the results of screening CE examination? If so, the algorithm should be clearly stated.

Reply: We appreciate your suggestion. Two patients who developed immune-checkpoint inhibitor-induced enterocolitis after capsule endoscopy had received colonoscopy first, referring to the results of capsule endoscopy at day 60. We described the process of deciding which site is to be examined first and corrected the sentences in the Result section “3.3 Summary of the two cases who developed ICI-induced colitis.” As mentioned, in this study, we were not able to regulate the decision whether to perform an esophagogastroduodenoscopy or colonoscopy after the capsule endoscopy showed any findings. Hence, this is one of the important limitations of this study, and we have added sentences referring to this shortcoming in the limitation section of the Discussion. In future investigations, we consider performing endoscopy after capsule endoscopy detects any findings, based on the severity of patients’ symptoms, such as CTCAE grade ≥ 2, as stated in the guidelines.

Result

Case 1

A 67-year-old man diagnosed with metastatic renal cell carcinoma received nivolumab after being refractory to axitinib. However, CCE-2 performed 60 days after nivolumab initiation showed erosions in the proximal colon, with no findings in the small intestine. On day 112, he was admitted to our hospital with grade 3 diarrhea. According to the CCE-2 results, colonoscopy was performed at first, which revealed the loss of vascular marking and an edematous mucosa at the transverse colon (Supplementary Figure 1A and B). He had no medication history associated with drug-induced colitis. Further, the fecal culture result was negative. The patient was intolerant to prednisolone administration, owing to grade 2 hiccups; hence, infliximab was administered, and the symptoms subsequently improved.

Case 2

A 50-year-old man diagnosed with gastric neuroendocrine carcinoma received pembrolizumab, owing to the high microsatellite instability status confirmed by polymerase chain reaction-based testing. On day 60, CCE-2 was performed, which revealed an obliterated area, erosion in the proximal colon, and moderate erosions in the distal colon. On day 120, the patient developed severe cough and diarrhea. CT revealed bilateral ground-glass opacity in the lungs; however, there were no findings in the gastrointestinal tract. Considering the CCE-2 results obtained on day 60, we performed colonoscopy, which showed slight and sporadic mucosal changes (Supplementary Figure 1C and D). Nonetheless, stool culture and Clostridium difficile toxin testing results were negative. The symptoms were subsequently evaluated as irAE Grade 2 and subsided following prednisolone administration.

Discussion, section of limitations

Fifth, the present study failed to regulate the decision whether to perform esophagogastroduodenoscopy or colonoscopy if capsule endoscopy revealed remarkable findings. Hence, we suggest performing endoscopic examination after capsule endoscopy detects any findings based on the severity of patients’ symptoms in future studies.